# Enantioselective acyl-trifluoromethylation of olefins by bulky thiazolium carbene catalysis

Sripati Jana [1], Matthew D. Wodrich [2] & Nicolai Cramer [1] ✉

Enantioenriched α-chiral β-fluorinated ketones are valuable structural motifs with application in several fields. The recently emerged concept of NHC-catalyzed radical acyl-trifluoromethylation of olefins offers a rapid route to construct racemic β-fluorinated ketones in a single step. Due to the lack of competent chiral NHC catalysts constructing these molecules in an enantio-selective manner remains an unmet challenge. Herein, we report a family of chiral thiazolium carbenes having bulky chiral flanking groups and offering three distinct positions with broad steric and electronic tunability. The catalysts display so far unmatched enantioselectivities for acyl-trifluoromethylations of simple unactivated olefins with a wide variety of aldehydes and Togni's reagent. The method provides a variety of enantioen-riched β-trifluoromethylated α-chiral ketones in high yields and excellent enantioselectivities up to 98:2 er. A potential applicability of this methodology is demonstrated through enantio- and diastereoselective late-stage functio-nalizations of pharmaceutical compounds.

Chiral ketones bearing an α-stereogenic center are important struc-tural motifs found in many biologically active compounds, natural products, pharmaceuticals, agrochemicals, and functional materials (Fig. 1a)[1–3]. The replacement of the C-H bonds by C-F bonds is an important pillar in drug discovery. Such substitution can significantly improve the molecule's physical and biological properties[4]. The tri-fluoromethyl group is particularly noteworthy for its ability to enhance binding selectivity, metabolic stability, membrane permeability and lipophilicity[5–10]. The synthetic value of α-chiral ketones paired with the benefits of incorporating a trifluoromethyl group renders the devel-opment of advanced synthetic methods to access enantioenriched β-trifluoromethylated α-chiral ketones highly attractive[11–13]. β-Tri-fluoromethylated α-chiral ketones may not only serve as biologically active intermediates[14,15] but can also be transformed into more com-plex chiral compounds due to the extensive transformability of the carbonyl group.

Due to the synthetic value of chiral α-stereogenic ketones, several strategies have been developed to access them (Fig. 1b)[16–28]. Fre-quently, these methods require rationally designed pre-functionalized starting materials compromising utility and sustainability. Current

available methods to enantioenriched fluorinated ketones are limited to the synthesis of α-trifluoromethyl substituted carbonyl compounds[29–33]. In contrast, strategies for homologous β-tri-fluoromethyl ketones containing an α-stereogenic center remain lar-gely elusive despite the numerous synthetic routes available for constructing enantioenriched α-chiral ketones. Synthetic procedures to access racemic β-trifluoromethylated α-branched ketones have been developed over the past decades. These methods are primarily based on either transition-metal catalysis, photoredox catalysis, or electrochemically generated radical-mediated functional group migration or intermolecular radical cascade reactions of tailor-made carbonyl-functionalized alkene substrates (Fig. 1c)[34–37].

Pioneered by Ohmiya and Studer, N-heterocyclic carbene (NHC) based radical catalysis showed remarkable synthetic versatility in the construction of carbon-carbon bonds due to capitalizing on their umpolung reactivity, the strong reducing power of the enolate inter-mediate, and the generation of persistent ketyl radicals[38–54]. Along these lines, Li[50] and Wang[51] reported a NHC-catalyzed dicarbo-functionalization of olefins using aldehydes and Togni's reagent as a trifluoromethyl source to construct racemic β-trifluoromethylated

[1]Laboratory of Asymmetric Catalysis and Synthesis, Institute of Chemical Sciences and Engineering, Ecole Polytechnique Fédérale de Lausanne (EPFL), Lausanne, Switzerland. [2]Computational Molecular Design Laboratory, Institute of Chemical Sciences and Engineering, Ecole Polytechnique Fédérale de Lausanne (EPFL), Lausanne, Switzerland. ✉e-mail: nicolai.cramer@epfl.ch

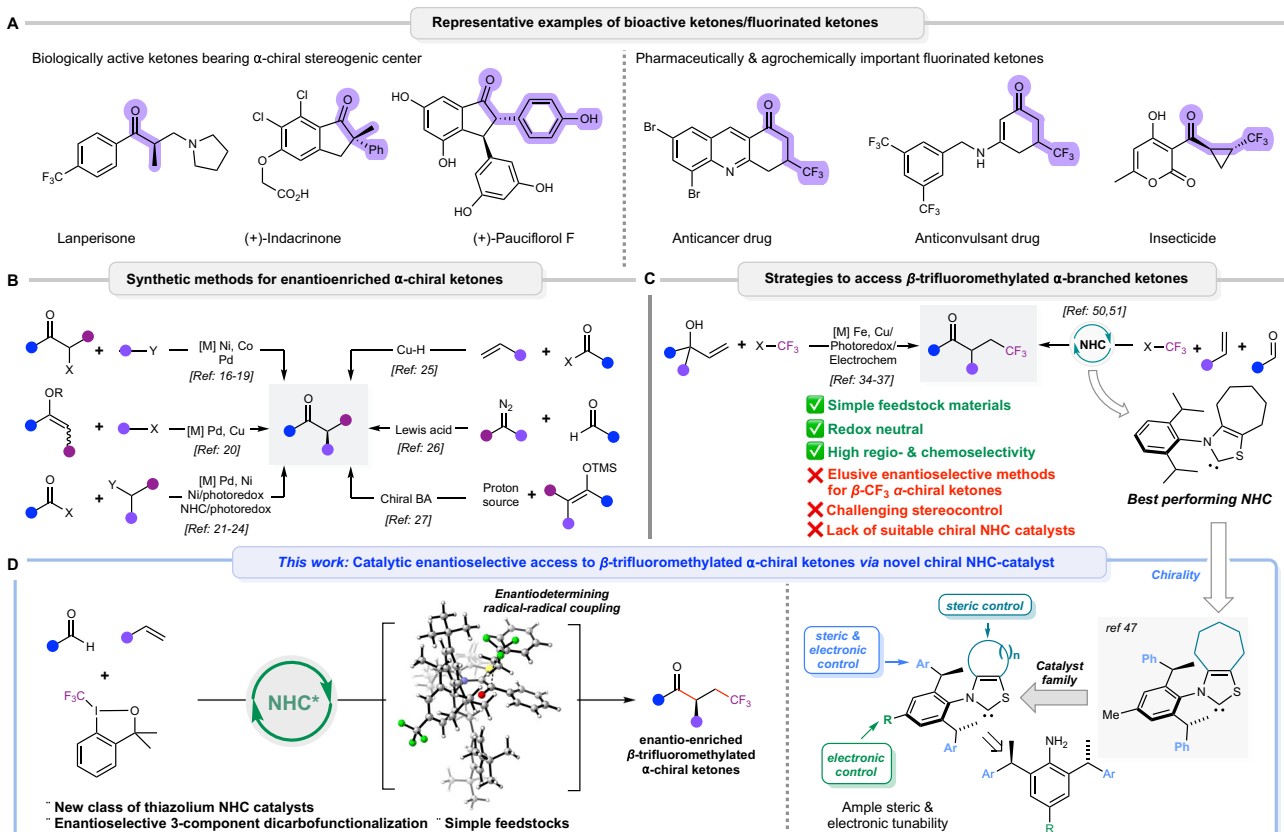

**Fig. 1 | Enantioselective synthesis of β-trifluoromethylated α-chiral ketones.**
**A** Biologically active compounds bearing α-stereogenic centers and pharmaceuticals and agrochemical with β-trifluoromethylated ketone motifs. **B** Synthetic methods for enantioselective synthesis of α-chiral ketones. **C** Strategies to access racemic β-trifluoromethylated α-branched ketones. **D** Highly-tunable thiazolium carbene catalysts enable an efficient enantioselective route to enantioenriched β-trifluoromethylated α-chiral ketones.

α-branched ketones. Noteworthy, Li conducted an extensive screening of numerous reported chiral NHCs for this transformation to achieve a chiral induction. Despite these substantial optimization efforts, the best-performing chiral NHC provided only a poor asymmetric induction of 20% ee. Contrasting the advancements in racemic dicarbofunctionalization of olefins via NHC radical catalysis[49–54], enantioselective transformations remain unsolved challenges. The lack of success in large screening efforts of known chiral NHCs clearly points to a significant gap in the toolbox requiring the design and development of new chiral NHC ligands able to overcome the selectivity shortcomings. We envisioned that a suitably modifiable designer chiral thiazolium carbene could efficiently engage in enantioselective NHC-catalyzed radical-dicarbofunctionalizations. The relevance of enantioenriched β-trifluoromethylated α-chiral ketones in various domains[34–36,55,56] makes the transformation of simple olefins with aldehydes and trifluoromethyl radical precursors an ideal testbed for this hypothesis (Fig. 1d).

In this work, we report a class of highly tunable chiral thiazolium carbenes that display high efficiency in the enantioselective three-component radical acyl-trifluoromethylation reaction, affording a diverse range of enantioenriched β-trifluoromethylated α-chiral ketones in high yields and enantioselectivities.

## Results

### Catalyst development and reaction optimization

We reasoned that our chiral $C_2$-symmetric 2,6-di-(1-arylethyl)aniline platform that previously served as key foundation for tunable chiral imidazolium IPr-type NHC ligand families[57], particularly successful for enantioselective Ni-catalyzed transformations[58–62] could serve as the chiral element of the envisioned thiazolium carbenes. This thiazolium carbene family is highly tunable (Fig. 1d). The substituent R offers electronic control on the carbene carbon atom. The flanking aryl groups of the chiral selector as well as the fused aliphatic backbone ring provide a finely adjustable steric control. The synthetic assembly of the thiazolium salt follows either the reported synthesis[63] for 2,6-diisopropylaniline-derived thiazolium salts (**NHC1-NHC4**) or a modified route for bulkier members (**NHC5-NHC7**) (see SI for details). Li previously reported a simple member of this NHC catalyst family for an intramolecular C-H acylation reaction[47] and most recently applied the same catalyst to a 1,2-boron migrative acylation, again observing a moderate enantiomeric excess[64]. Given our expertize in the systematic engineering of chiral $C_2$-symmetric 2,6-di-(1-arylethyl)aniline platform by incorporating multiple steric and electronic adjustments to improve enantioselectivity, we subsequently tested the suitability of the developed chiral thiazolium salts for the enantioselective three-component dicarbofunctionalization reaction to access β-trifluoromethylated α-chiral ketones. Heating benzaldehyde and styrene as the model substrate combination in the presence of pre-catalyst **NHC***, Togni I reagent (**F1**) and $Cs_2CO_3$ at 60 °C provided targeted ketone **3a** (Table 1). **NHC1** proved to be an active catalyst providing ketone **3a** in 71% isolated yield and 60:40 er (Entry 1). Next, we systematically investigated how structural modifications to the catalyst—specifically, substitutions altering steric and electronic profiles—influence reactivity and selectivity outcomes. When an electron-rich substituent (-OMe) was introduced at the *para*-position of the aniline nitrogen (**NHC2**), neither reaction efficiency (yield) nor enantioselectivity (er) showed meaningful improvements (Entry 2). In contrast, installing an electron-deficient *para*-trifluoromethyl group (-CF₃,

**Table 1 | Optimization of the asymmetric synthesis fluorinated ketone[a]**

| Entry | NHC* | Changes | % Yield 3a[c] | er[d] |
|---|---|---|---|---|
| 1 | NHC1 | - | 73 (71)[e] | 60:40 |
| 2 | NHC2 | - | 74 (73)[e] | 60:40 |
| 3 | NHC3 | - | 64 | 67:33 |
| 4 | NHC4 | - | 60 | 70:30 |
| 5 | NHC5 | - | 58 | 82:18 |
| 6 | NHC6 | - | 59 | 85:15 |
| 7 | NHC7 | - | 55 | 88:12 |
| 8 | NHC8 | - | 58 | 60:40 |
| 9 | NHC9 | - | 39 | 56:44 |
| 10 | NHC7 | TBME as solvent | 60 | 91:9 |
| 11 | NHC7 | TBME as solvent, 40 °C | 39 | 96:4 |
| 12[b] | NHC7 (10 mol%) | TBME as solvent, 40 °C, 36 h | 79 (76)[e] | 96:4 |
| 13 | NHC7 | F2 instead of F1 | 19 | 95:5 |
| 14 | NHC8 | F3 instead of F1 | <5 | - |

[a] Conditions: 0.15 mmol 1a, 0.1 mmol 2a, 0.2 mmol F1, 5 μmol NHC*, 15 μmol Cs₂CO₃ in 1 mL DCM at 60 °C for 18 h. [b] 10 μmol NHC7 and 25 μmol Cs₂CO₃. [c] ¹H NMR yield using 1,3,5-trimethoxybenzene as internal standard. [d] Determined by chiral HPLC. [e] Isolated yield. TBME - Methyl tert-butyl ether.

**NHC3**) significantly enhanced stereocontrol, delivering **3a** in 64% yield with an er of 67:33 (Entry 3). To disentangle steric contributions from electronic effects, we next varied the aliphatic ring size within the thiazolium backbone. Expanding the saturated ring to an eight-membered system (**NHC4**) further improved enantioselectivity to 70:30 er (Entry 4), reinforcing the role of steric bulk in asymmetric induction. Collectively, these results indicate that electronic tuning (via electron-withdrawing groups) and steric modulation (via back-bone enlargement) independently elevate enantioselectivity. To exploit this synergy, we designed a catalyst (**NHC5**) merging a *para*-CF$_3$-substituted aniline with a twelve-membered saturated backbone ring. This dual modification dramatically improved stereochemical outcomes, achieving an er of 82:18 for **3a** (58% yield, Entry 5). Finally, we explored whether introducing additional steric hindrance at the chiral side arm adjacent to the aniline motif could further refine enantioselectivity. Indeed, increasing the steric demand of the aryl side arms (**NHC6**, Ar = *m*-xylyl; **NHC7**, Ar = 3,5-*t*Bu-C$_6$H$_3$) led to a further improvement of enantioselectivity (85:15 and 88:12 er, respectively) while maintaining the reaction yield (Entries 6, 7). The well-established catalysts from Rovis (**NHC8**) and Sheehan (**NHC9**) were not well suited providing **3a** in moderate yield and very modest enantioselectivity (Entries 8-9). These results are in line with the previous observations[50]. Further optimization involved screening solvents, bases, reaction temperatures and times, catalyst loadings, and stoichiometries of the reaction components (see SI). As result, conducting the reaction in *tert*-butyl methyl ether (TBME) as solvent provided **3a** in a slightly improved yield (60%) and enantioselectivity (91:9 er) (Entry 10). Reducing the reaction temperature to 40 °C increased the selectivity to 96:4 er (Entry 11). The reaction yield was increased to 76% combining a higher catalyst loading with prolonged reaction time (Entry 12). Alternative trifluoromethyl radical precrsors such as Togni II (**F2**) and Umemoto reagent (**F3**) led to a very low yield or no formation of **3a** (Entries 13-14).

## Substrate scope

With the optimized reaction conditions, we explored the applicability of this method with a broader substrate scope (Fig. 2). A wide range of aliphatic and aromatic aldehydes were well tolerated and furnished the corresponding $\beta$-trifluoromethylated $\alpha$-chiral ketones (**3a**-**3o**) in high to excellent yields and excellent enantioselectivities. Aromatic alde-hydes with electron-withdrawing and electron-donating groups were well tolerated and allowed isolation of corresponding products **3b**-**3e** in good to high yields and excellent enantioselectivities. Aromatic aldehydes with electron-donating groups (**1b** and **1e**) displayed slightly higher enantioselectivities than those with electron-withdrawing sub-stituents (**1c** and **1 d**). This effect might be attributed to a very slow racemization of the $\alpha$-chiral center in the presence of base Cs$_2$CO$_3$ in the reaction medium for substrates with electron-withdrawing groups. The absolute configuration of the trifluoromethylated ketones was determined to be *S* by single crystal X-ray crystallographic analysis of compound **3 d**[65]. (CCDC 2360496 (**3d**) contains the supplementary crystallo-graphic data for this paper. These data are provided free of charge by the Cambridge Crystallographic Data Centre.) Sterically demanding *ortho*-substituted aldehydes **1 f** and **1 g** were well tolerated the reaction conditions. Of particular interest, 2-hydroxybenzaldehyde (**1 g**) reacted selectively at the aldehyde site, yielding the desired product **3 g** in high yield and excellent enantioselectivity, without any byproduct formation from the reactive hydroxyl group. Both electron-rich and electron-poor heteroaromatic aldehydes (**1i-1l**) also compe-tently engaged in the transformation providing products **3i-3l** in high yields and high to excellent enantioselectivities. The scope of the dicarbofunctionalization reaction can be extended to aliphatic alde-hydes. For example, aldehydes **1m-1o** were competent and reacted with comparable reactivity and selectivity to those of aromatic alde-hydes, resulting in the formation of products **3m-3o**. Noteworthy,

cyclopropanecarbaldehyde **1o** exclusively delivered cyclopropyl-bearing product **3o**. No ring-opened products that typically occurs on transformations involving cyclopropylmethyl radicals. Based on the reported mechanism[49–54] and a study on the critical assessment of the reducing ability of the Breslow enolate intermediate[65], we speculate that the ketyl radical intermediate **3o-1** indeed forms. However, the extensive mesomeric delocalization (**3o-2** and **3o-3**) stabilized the radical and substantially attenuating its radical-clock type opening propensity. Next, the suitability of different olefins was explored. Styrenes with electron-donating groups reacted smoothly and affor-ded trifluoromethylated ketones **4a, 4b**, and **4 d** in high yields and excellent enantioselectivities. *p*-Cyano styrene reacted well giving the corresponding product **4c** in 62% yield and slightly diminished enan-tioselectivity, again attributable to the decreased pKa value of the $\alpha$-proton and slow partial racemization under the basic reaction condi-tions. Different heteroaryl-substituted olefins efficiently participated in the transformation delivering trifluoromethylated ketones **4e-4g** in excellent enantioselectivities. To further demonstrate the robustness and generality of this method, we tested an array of non-conjugated alkenes. Heteroatom-substituted (*S, N*) non-conjugated olefins, such as phenyl vinyl sulfide (**2 h**) and vinyl phthalimide (**2i**), underwent radical acyl trifluoromethylation competently, delivering the desired pro-ducts (**4 h, 4i**) in good yields and excellent enantioselectivity. Unbiased terminal aliphatic alkenes **2j** and **2k** offering less stabilization to the transient 2° radical reacted to products **4j** and **4k** in slightly reduced yield but with similarly high enantio-induction. Additionally, we examined the scope of Togni I-type reagents substituted with perfluoroalkyl groups. Both heptafluoropropyl- and nonafluorobutyl-substituted hypervalent iodine reagents reacted efficiently, yielding the corresponding perfluoroalkylated $\alpha$-chiral ketones **4 l** and **4 m** in high yields and with excellent enantioselectivities.

## Application to pharmaceutically relevant molecules

To showcase further synthetic utility, we demonstrated late-stage enantioselective acyl-trifluoromethylations of pharmaceutically rele-vant compounds[66–68]. The vinyl derivative of fenofibrate **5a**, clinically used for hyperlipemia treatment[69], was subjected to the reaction conditions providing acyl trifluoromethylated modified fenofibrate **6a** in good yield and with high asymmetric induction. The method is as well suitable for the selective modification of chiral drugs such as olefinic derivatives of estrone and $\delta$-tocopherol, both featuring mul-tiple chiral centers. The corresponding products **6b** and **6c** were formed in high yields and excellent diastereoselectivities. The observed diastereoselectivities are strong evidence for a full catalyst control underlining the critical relevance of the chiral NHC catalyst in the enantio-determining C-C bond forming event of the radical-radical recombination.

## Control experiments

To confirm the step at which chirality is introduced in this reaction and to evaluate the potential for epimerization of enantioenriched $\beta$-tri-fluoromethylated $\alpha$-chiral ketones under basic conditions, we con-ducted a series of control experiments (Fig. 3a). First, 92% enantiopure **3a** was treated with 25 mol% Cs$_2$CO$_3$ in TBME (0.1 M) at 40 °C for 16 hours, resulting in only a 1% decrease in enantiopurity. Next, we exposed 92% enantiopure **3a** to either 10 mol% **NHC7** or a combination of 10 mol% **NHC7** and 25 mol% Cs$_2$CO$_3$ in TBME (0.1 M) at 40 °C for 16 hours, and in both cases, complete retention of enantiopurity was observed. Furthermore, heating a *rac*-**3a** in the presence of 10 mol% **NHC7** and 25 mol% Cs$_2$CO$_3$ in TBME (0.1 M) at 40 °C for 16 hours showed no evidence of chiral induction, ruling out any possibility of chirality construction via asymmetric protonation. These findings strongly suggest that, following product formation, the $\alpha$-chiral center is not subject to epimerization induced by either Brønsted basic **NHC7′** or Cs$_2$CO$_3$, nor does asymmetric protonation contribute to chirality.

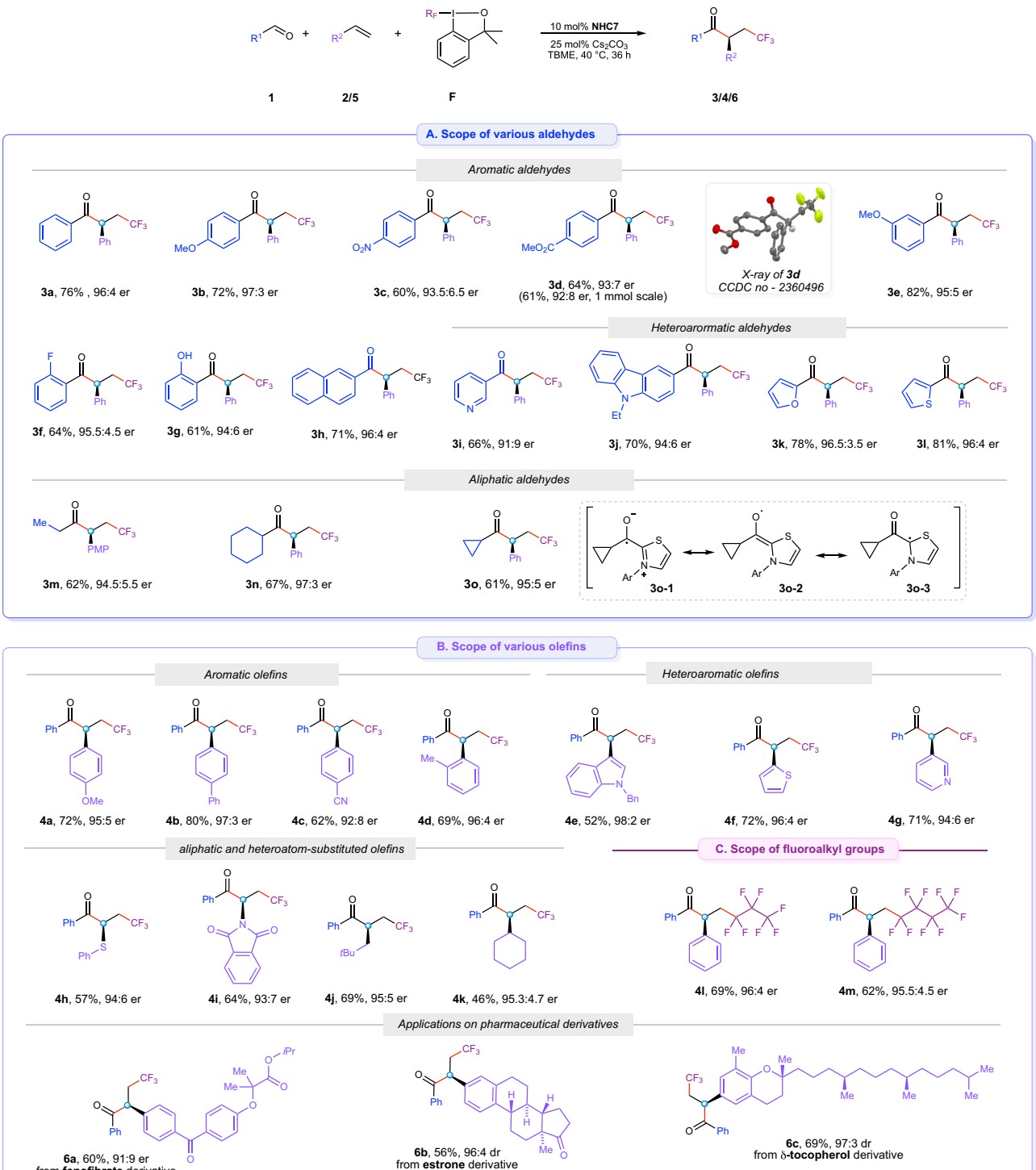

**Fig. 2 | Scope for asymmetric acyl-trifluoromethylation reaction. A** Scope of various aldehydes. **B** Scope for various olefins. **C** Scope for various hypervalent iodine reagents. Isolated yields. Enantiomeric ratio determined by chiral HPLC.

We thus conclude that the observed enantioselectivity arises directly from the radical-radical coupling step, effectively ruling out significant epimerization at the chiral α-center under the reaction conditions.

## Reaction mechanism and computational analysis

Based on the results obtained from the control experiments and literature precedents[23,24,49–54] a putative catalytic cycle for the thiazolium carbene-catalyzed enantioselective dicarbofunctionalization reactions of simple olefins can be formulated (Fig. 3b). The catalytic cycle begins

with base-assisted generation of Breslow enolate **C** from **NHC A** and aldehyde **B**. Single electron transfer from **C** to the **F1** reagent then produces a transient trifluoromethyl radical alongside the stabilized ketyl radical **D**. The trifluoromethyl radical undergoes addition to the olefin, yielding secondary radical **E**. Enantioselectivity arises from the coupling of radicals **D** and **E**, a C-C bond-forming process that also likely constitutes the rate-determining step. The facile extrusion of carbene **A** delivers enantioenriched β-trifluoromethylated α-chiral ketone **3** closing the catalytic cycle. To understand the basis for

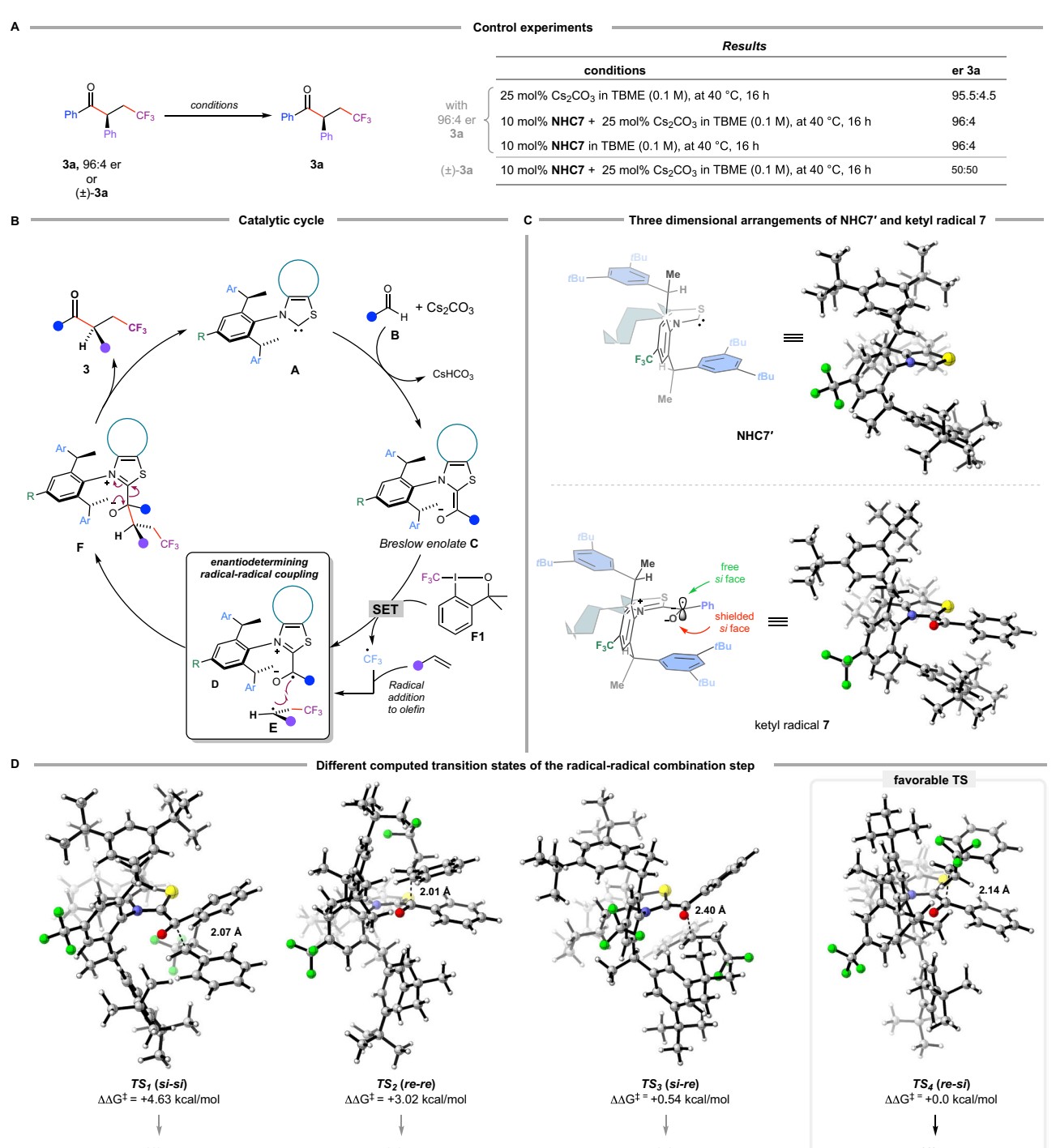

**Fig. 3 | Mechanism study. A** Control experiments. **B** Suggested catalytic cycle. **C** Three-dimensional arrangements of **NHC7′** and ketyl radical intermediate **7**. **D** Different computed transition states of the radical-radical combination step (Free energies determined at the PBE0-D3(BJ)/def2-TZVP//PBE0-D3(BJ)/def2-SVP level).

enantioselectivity, we propose an asymmetric induction model for the radical-radical coupling step using benzaldehyde **1a**, styrene **2a**, reagent **F1** and **NHC7′**. We first determined geometries corresponding to the global minima of **NHC7′** (Supplementary Data 1) and its ketyl radical intermediate **7** (Supplementary Data 2), as well as the transition states leading to the major and minor enantiomer of the product using computational chemistry. To accomplish this, we employed a protocol that relied on rapid, yet extensive conformer screening using Grimme's conformer-rotamer ensemble sampling tool (CREST)[70]. Of the many thousands of conformers identified, the ten most representative conformers (for each structure/TS) lying within 15 kcal/mol

of the global minima assigned by CREST were selected using *marc*, a conformer analysis program[71]. The geometries and free energies of these species were then determined using DFT computations at the PBE0[72,73]-D3(BJ)[74,75]/def2-TZVP[76]//PBE0-D3(BJ)/def2-SVP[76] level including solvation corrections using the SMD model[77] in Gaussian16[78] (for full details see SI). Examining the energetics of the four distinct enantio-determining transition states, we found that the *si-si* (**TS₁**, Supplementary Data 3) and *re-re* (**TS₂**, Supplementary Data 4) interactions leading to the (S)- and (R)-enantiomers of product **3a** to both be energetically unfavorable (see Fig. 3). The relevant pathways to the observed products occur through either a *si-re* interaction (**TS₃**,

Supplementary Data 5) that leads to the minor (*R*)-enantiomer ($\Delta\Delta G^{\ddagger}$ = +0.54 kcal/mol, Fig. 3) or a *re-si* interaction (**TS₄**, Supplementary Data 6) that leads to the major (*S*)-enantiomer ($\Delta\Delta G^{\ddagger}$ = 0.00 kcal/mol, Fig. 3). Conversion of these Gibbs free energy differences into a theoretical er obtained through Boltzmann weighting of all four transition states yields a *S:R* ratio of 80:20, which aligns well with the experimental results.

## Discussion

In conclusion, we report a family of chiral thiazolium carbenes and showcased their efficiency and selectivity in a rare example of enantioselective radical NHC catalysis. This thiazolium carbene class has bulky chiral flanking groups and offers three distinct positions with broad steric and electronic tunability. The catalysts display so far unmatched enantioselectivities for acyl-trifluoromethylations of simple unactivated olefins with a wide variety of aldehydes and Togni's reagent. This method provides an efficient and straightforward route to access a broad range of highly valuable enantioenriched $\beta$-trifluoromethylated $\alpha$-chiral ketones in high yields and excellent enantioselectivities. The synthetic utility was further demonstrated by the enantioselective late-stage functionalization of derivatives of pharmaceuticals. Further investigations leveraging this class of chiral NHC for enantioselective olefin dicarbofunctionalization are currently underway in our laboratory.

## Methods

### General procedure for the chiral thiazolium carbene-catalyzed asymmetric acyl-trifluoromethylation of olefins

In a nitrogen-filled glove box, an oven-dried 5 mL reaction tube was charged with chiral thiazolium salt **NHC7** (0.01 mmol, 10 mol%). Then, 1 mL of dry and degassed methyl *tert*-butyl ether (TBME) was added. Subsequently, aldehyde **1** (0.15 mmol, 1.5 eq.), olefin **2** or **5** (0.1 mmol, 1.0 eq.), Togni reagent (0.2 mmol, 2 eq.), and Cs₂CO₃ (0.025 mmol, 25 mol%) were added to the reaction tube, which was then sealed with a crimper. The reaction tube was removed from the glove box and stirred at 40 °C for 36 hours. The reaction mixture was then filtered through a short pad of silica gel (2 cm), and the solvent was evaporated under reduced pressure. The crude product mixture was purified by silica gel column chromatography using *n*-pentane: ethylacetate mixture as eluent to obtain corresponding enantioenriched $\beta$-trifluoromethylated $\alpha$-chiral ketones (**3/4/6**).

## Data availability

The authors declare that the data supporting the findings of this study are available within the paper and its Supplementary Information file. The experimental procedures and characterization of all new compounds are provided in the Supplementary Information. Data supporting the findings of this manuscript are also available from the corresponding author upon request. Crystallographic data for compound **3 d** has been deposited at Cambridge Crystallographic Data Center, under accession code CCDC no. 2360496. Copies of the data can be obtained free of charge via https://www.ccdc.cam.ac.uk/structures/. The CIF file of **3 d** is also included in the Supplementary Information file.

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

## Acknowledgements

This work is supported by the EPFL and the NCCR Catalysis. This publication was created as part of NCCR Catalysis (grant number 180544), a National Center of Competence in Research funded by the Swiss National Science Foundation. We thank Dr. R. Scopelliti for X-ray crystallographic analysis of compound **3 d**. The Laboratory for Computational Molecular Design at EPFL is acknowledged for providing computational resources.

## Author contributions

S.J. and N.C. conceived the project and designed all the experiments. S.J. performed the experiments. M.D.W. performed the computational studies. All the authors analyzed the experimental & computational results and co-wrote the manuscript.

## Competing interests

The authors declare no competing interests.
