## [Transparent Peer Review file · Nature Communications]

Enantioselective Acyl-Trifluoromethylation of Olefins by Bulky Thiazolium Carbene Catalysis

Corresponding Author: Professor Nicolai Cramer

Version 0:

Reviewer comments:

Reviewer #1

(Remarks to the Author)

The manuscript by Cram and Jana describes an NHC-organocatalyzed enantioselective radical acyltrifluoromethylation of styrenes. A class of chiral thiazolium catalyst is developed to achieve high ee. 28 examples of α -chiral β -fluorinated ketone products were showed and most of them were with good yields and excellent ee.

Although the same reaction has been developed by Li and Wang, the successful enantioselective control in this manuscript is surely of high importance.

In the design of chiral thiazolium catalysts which are widely used in asymmetric NHC organocatalysis, bulky N-substituents (eg. Mes) are normally needed to ensure a confined catalytic chiral pocket. However, in the case of thiazolium catalyst, no substituents can be introduced to the sulfur atom. As a result, it is believed that small molecule chiral thiazoliums can hardly work in asymmetric NHC organocatalysis.

In this manuscript, the author smartly introduced a 12-membered ring carbocycles to somehow provide enough steric hinders to achieve the enantioselective control. It does open new envisions in the development of chiral thiazolium catalyst. I support its publication in Nat. Commun.

Some of the minor problems should be revised as follows:

1. Besides acyltrifluoromethylation, acyldifluoromethylation was also viable in Li's work (ref 50). Some examples of enantioselective acyldifluoromethylation of styrenes can surely further enhance the significance of this manuscript.
2. Some of the ¹⁹F NMR spectrums (eg. 3c, 3d, 3e, 3f, 3k, 4c, 4e, 4f, 4i, 4j, 6a, 6b) are not clean enough and need further purification.
3. The abbreviations for methyl tertiary butyl ether should be uniform, e.g. MTBE or TBME.
4. Spaces were needed before and after "=", (eg. "R = OMe" rather than "R=OMe")
5. The HRMS data of compound 4f is not completed (PS42 in SI).

Reviewer #2

(Remarks to the Author)

In this manuscript, the authors present the NHC-catalyzed enantioselective acyl-trifluoromethylation of olefins with aldehydes and Togni's reagent. While the racemic version of this reaction has been previously reported, achieving effective chiral control has remained a challenge, with prior studies reporting only up to 20% ee (ref. 50). The authors have made significant progress by achieving up to 96% ee using a thiazolium NHC pre-catalyst incorporating the chiral framework of C₂-symmetric 2,6-di-(1-arylethyl)aniline. However, the introduction lacks a detailed discussion of the foundational work behind this class of chiral catalysts. Importantly, the Li group pioneered this category of chiral NHC catalysts and applied them in remote C-H acylation, achieving up to 50% ee (ref. 47). Zhao and co-workers also employed similar catalysts in the aminoacylation of olefins (JACS, 2022, 144, 22767). Other application includes a boron migrative acylation reaction (Sci. Adv. 2024, 10.1126/sciadv.adn8401). Although these studies did not reach highly enantioselective outcomes, it is important for the authors to provide a precise and comprehensive review of this background.

The main highlight of this article is the authors' successful optimization of the catalysts, leading to a significant improvement in enantioselectivity. This is a noteworthy result and a substantial contribution to the emerging field of radical NHC catalysis.

For these reasons, I believe the manuscript is suitable for publication in Nature Communications after the following revisions are addressed:

1. As noted, the introduction should include a more detailed account of the foundational contributions to the development of chiral NHC catalysts, including key work by the Li group and others. These description should added in both text and figures.
2. In Table 1, the size of the fused aliphatic ring appears to impact enantioselectivity significantly. It would be interesting to evaluate NHC catalysts with larger fused rings (>12 members).
3. The source of chirality in this reaction requires further verification. Although it appears that chirality is constructed during the radical coupling step, it's possible that the α -chirality of the carbonyl could racemize under the basic conditions of NHC catalysis. Since NHC catalysts can act both as Lewis bases and strong Brønsted bases, this raises the possibility that the chirality could be racemized and rebuilt through asymmetric protonation by the carbene as a chiral base. More mechanistic experiments should be conducted to rule out this pathway.
4. The substrate scope should be expanded to include o-substituted aryl aldehydes, enals and fluoroalkylated reagents beyond CF₃.
5. The structure of the ketone in Figure 1A should be revised.
6. In Table 1 and Tables S1–S6, ensure consistent reporting of yields (NMR yields vs. isolated yields) for clarity.
7. In Figures 2B and 2C, the energy gap between the Si-Re and Si-Si transition states should be provided.
8. In the supporting information (SI), some HPLC spectra (e.g., for compounds 4f and 6a) show impurities. Please review and ensure the purity of all spectra.
9. The internal standard for all ¹⁹F NMR spectra in the SI should be included.

Version 1:

Reviewer comments:

Reviewer #1

(Remarks to the Author)

The point-to-point respond is satisfactory.
The recent version is suitable for publication.

Reviewer #2

(Remarks to the Author)

Most of my previous concerns have been well-addressed by the authors, with the exception for the following issues:

1. For the comment 2 about the impact of fused ring on catalysts

The article's innovation in catalyst modification is commendable. However, the investigation into how various fused aliphatic rings influence enantioselectivity remains incomplete. While the authors have examined the impact of 8-membered and 12-membered rings, the effects of other ring sizes, such as 10-membered and 14-membered rings, on both reaction conversion and enantioselectivity remains to be elucidated to guide the catalyst design. In their response, the authors's statement "it seems that the ring size effects reach saturation at some point" lacks experimental support. Therefore, a more comprehensive investigation into the role of these ring sizes in chiral catalysts is necessary.

2. For the comment 7 about the calculation study

In light of the author's explanation, I can understand why the author did not calculate the energy gap between different transition states for the enantio-determining step. However, the DFT results presented in the current paper are somewhat rudimentary. Despite the vast conformational space of the organocatalyst and its catalytic intermediates, there is no mention of any attempts to verify the assignment of key structures as global conformational minimum. The process for geometry optimization for the NHC catalyst and the radical intermediates should be detailed. In addition, considering the solvent significantly impacts the reaction, the presented calculation study, which was conducted in vacuo, should ideally incorporate solvent effects for enhanced accuracy.

3. We highly suggest the author should clarify this catalyst design from Li's group before ref. 47 in Figure 1D.

Version 2:

Reviewer comments:

Reviewer #3

(Remarks to the Author)

The authors have well answered the questions, and managed to provide detailed DFT calculations to explain for the enantioselectivity generated. However, one key question is still remained to be uncertain: whether the reaction mechanism is a radical chain process or not. A radical-radical C-C coupling is not kinetically favored because of the low concentration of highly-reactive radical species. The benzyl radical E may attack the aldehyde 1a, which is in a higher concentration, rather than the radical anion D. This is a shared challenge for both the present work and Li's non-enantioselective work. I think the answer to this question will help a lot in this field. Therefore, major revision is still needed

POINT-BY-POINT RESPONSE TO REVIEWER COMMENTS

Reviewer #1:

Comment 1. Besides acyltrifluoromethylation, acyldifluoromethylation was also viable in Li's work (ref 50). Some examples of enantioselective acyldifluoromethylation of styrenes can surely further enhance the significance of this manuscript.

Our answer: We appreciate the reviewer's thoughtful suggestion regarding the inclusion of enantioselective acyldifluoromethylation. However, we would like to clarify that incorporating acyldifluoromethylation of styrene requires another class of radical precursors, distinct from the hypervalent iodine Togni reagent reported here and hence different sets of optimization and specific reaction conditions. Our current manuscript focuses on developing a sustainable synthetic approach to enantioenriched β -trifluoromethylated α -chiral ketones (as well as in the revision perfluoroalkyl analogues), using newly developed chiral thiazolium carbenes and hypervalent iodine reagents.

The acyl-difluoroalkylation reaction indeed represents a promising area that clearly merits independent attention, as it offers potential for accessing a broad spectrum of β -difluoro-substituted α -chiral ketones with varied R groups, including difluorobromoesters, difluorobromoamides, difluorobromosulfones. Given the distinctive reactivity and potential applications of these molecules, we believe a focused investigation on acyldifluoroalkylation would better serve the scientific community as a separate study rather than being an easily overlookable side-kick in this manuscript. We believe maintaining a sharp focus on β -trifluoromethyl and perfluoroalkyl-substituted ketone synthesis allows for a clearer and more concise presentation of our current findings.

Comment 2. Some of the ^{19}F NMR spectrums (eg. 3c, 3d, 3e, 3f, 3k, 4c, 4e, 4f, 4i, 4j, 6a, 6b) are not clean enough and need further purification.

Our answer: We have re-purified the underlying compounds and all the ^{19}F NMR spectra (e.g., 3c, 3d, 3e, 3f, 3k, 4c, 4e, 4f, 4i, 4j, 6a, and 6b) are now clean and replaced the old spectra. *Please note in the revised version of the manuscript and SI compound 3f is 3h, 3k is 3m.*

Comment 3. The abbreviations for methyl tertiary butyl ether should be uniform, e.g. MTBE or TBME.

Our answer: We have made the correction and are using TBME as the abbreviation for tert-butyl methyl ether.

Comment 4. Spaces were needed before and after "=", (eg. "R = OMe" rather than "R=OMe")

Our answer: We have made the correction.

Comment 5. The HRMS data of compound 4f is not completed (PS42 in SI).

Our answer: We thank the reviewer for spotting this mistake. We have corrected the HRMS data for compound 4f in the supporting information.

Reviewer #2 (Remarks to the Author):

Comment 1. As noted, the introduction should include a more detailed account of the foundational contributions to the development of chiral NHC catalysts, including key work by the Li group and others. These description should added in both text and figures.

Our answer: We appreciate the reviewer's feedback and acknowledge the foundational contributions of Li et al. in chiral NHC catalysis. We have added their contributions to the text and scheme. They investigated one single member of the NHC catalyst using the simplest chiral aniline for an *intramolecular* C-H acylation with some 50% ee. In addition, during the preparation of our manuscript, Li et al. reported the same catalyst for a single substrate in 1,2-boron migrative acylation with modest 50% ee.

Our study aims for the asymmetric synthesis of α -chiral β -trifluoromethylated ketones via NHC-catalyzed radical-radical coupling, with a strong emphasis on enantioselectivity and catalyst design. The introduction of our

manuscript reflects this goal by discussing relevant synthetic strategies and referencing the work of Li and Wang to provide context for our catalyst modifications. Our successful design strategy provides innovation to enhance enantioselectivity by introducing both a larger aliphatic backbone ring and steric chiral aniline modulations to create a more confined chiral pocket. In response to the reviewer's comments, we added a discussion of the reports by Li et al. and accordingly revised Figure 1 and cited the new literature (Sci. Adv. 2024, 10.1126/sciadv.adn8401) in the text.

Comment 2. In Table 1, the size of the fused aliphatic ring appears to impact enantioselectivity significantly. It would be interesting to evaluate NHC catalysts with larger fused rings (>12 members).

Our answer: We appreciate the reviewer's suggestion regarding the evaluation of NHC catalysts with larger fused rings (>12 members). While we considered this during our optimization, the results obtained with the 12-member ring were meeting our cut-off criteria for the optimization. In addition, it seems that the ring size effects reaching saturation at some point.

Comment 3. The source of chirality in this reaction requires further verification. Although it appears that chirality is constructed during the radical coupling step, it's possible that the α -chirality of the carbonyl could racemize under the basic conditions of NHC catalysis. Since NHC catalysts can act both as Lewis bases and strong Brønsted bases, this raises the possibility that the chirality could be racemized and rebuilt through asymmetric protonation by the carbene as a chiral base. More mechanistic experiments should be conducted to rule out this pathway.

Our answer: We appreciate the reviewer's insightful suggestion to further verify the source of chirality in our reaction. To investigate the possibility of enantioselectivity arising from racemization followed by asymmetric protonation via the NHC catalyst, we additionally performed several control experiments and added them to the manuscript:

Control Experiment 1: A sample of **3a** (92 % ee) was exposed to 25 mol% Cs₂CO₃ in TBME (0.1 M) at 40 °C for 16 hours. Subsequent chiral HPLC analysis gave **3a** (91 % ee) - a marginal erosion in enantiopurity.

Control Experiment 2: We subjected **3a** (92 % ee) to the exact reaction conditions: 10 mol% chiral thiazolium salt (NHC7) with 25 mol% Cs₂CO₃ in TBME (0.1 M) at 40 °C for 16 hours. Again, chiral HPLC analysis showed negligible erosion of enantiopurity.

Control Experiment 3: We exposed **3a** (92 % ee) to 10 mol% NHC7 in TBME (0.1 M) at 40 °C for 16 hours. Chiral HPLC analysis confirmed complete retention of enantiopurity.

Control experiment 4: A racemic mixture of **3a** was subjected to exact reaction conditions: 10 mol% chiral thiazolium salt (NHC7) with 25 mol% Cs₂CO₃ in TBME (0.1 M) at 40 °C for 16 hours. Chiral HPLC analysis confirmed no chiral induction in the racemic mixture of **3a**, ruling out the possibility of chirality construction via asymmetric protonation.

These findings strongly indicate that, following product formation, there is no epimerization of the chiral center facilitated by the NHC catalyst, nor is there any reconstruction of chirality through asymmetric protonation. Based on these results, we conclude that the observed enantioselectivity originates directly from the radical-radical coupling step, effectively ruling out any significant potential for epimerization at the chiral α -center under the reaction conditions. These data have been included in the manuscript text and in Figure 2. Experiments description have been included in the SI.

Comment 4. The substrate scope should be expanded to include *o*-substituted aryl aldehydes, enals and fluoroalkylated reagents beyond CF₃.

Our answer: Regarding *o*-substituted aryl aldehydes, we added two new examples in the revised manuscript (**3f** and **3g**). To address the scope with enals, we investigated the reaction of *trans*-cinnamaldehyde. However, no product formation was observed. The majority of cinnamaldehyde (84%) remained intact in the reaction medium, indicating that the present catalytic system is not compatible with enals derivatives. For fluoroalkylated reagents beyond CF₃, we investigated C₃F₇- and C₄F₉-substituted hypervalent iodine reagent, and the results have been included in the manuscript (**4l** and **4m**).

Comment 5. The structure of the ketone in Figure 1A should be revised.

Our answer: We appreciate the reviewer for pointing out this mistake. We have revised and corrected all the ketone structures in Figure 1A.

Comment 6. In Table 1 and Tables S1–S6, ensure consistent reporting of yields (NMR yields vs. isolated yields) for clarity.

Our answer: We have added NMR yields and ensured consistent reporting of NMR versus isolated yields across Table 1 and Tables S1–S6 for clarity.

Comment 7. In Figures 2B and 2C, the energy gap between the Si-Re and Si-Si transition states should be provided.

Our answer: In this study, we focused on geometry optimization of the active NHC and the key ketyl radical intermediate to understand the 3D arrangement of the ketyl radical species during the proposed enantiodetermining radical-radical coupling step. Based on these optimized structures, we propose the Si-Si face attack as the more feasible pathway, with the Si-Re face attack being less favorable. We have not calculated the different transition state energies for a specific. Given the very high flexibility and large size of the catalyst it is very challenging to obtain accurate enough values (namely complication by the dispersion correction) that would have meaningful scientific underlying and not just two possible TSs with right 1.5 something kcal/mol to match the observed er's and provide a false impression of accuracy.

Comment 8. In the supporting information (SI), some HPLC spectra (e.g., for compounds 4f and 6a) show impurities. Please review and ensure the purity of all spectra.

Our answer: We appreciate this comment. In the revised SI, we have added improved HPLC spectra for compounds 3c, 3e, 4c, 4f, and 6a.

Comment 9. The internal standard for all ^{19}F NMR spectra in the SI should be included.

Our answer: We used CFCl_3 as the reference and have corrected all the ^{19}F spectra accordingly in the revised SI.

POINT-BY-POINT RESPONSE TO REVIEWER COMMENTS

Reviewer #1:

The point-to-point respond is satisfactory.

The recent version is suitable for publication.

Our comment: We thank the reviewer for their positive evaluation of our manuscript and are pleased that they are satisfied with our revision.

Reviewer #2:

Comment 1. For the comment 2 about the impact of fused ring on catalysts the article's innovation in catalyst modification is commendable. However, the investigation into how various fused aliphatic rings influence enantioselectivity remains incomplete. While the authors have examined the impact of 8-membered and 12-membered rings, the effects of other ring sizes, such as 10-membered and 14-membered rings, on both reaction conversion and enantioselectivity remains to be elucidated to guide the catalyst design. In their response, the authors's statement "it seems that the ring size effects reach saturation at some point" lacks experimental support. Therefore, a more comprehensive investigation into the role of these ring sizes in chiral catalysts is necessary.

Our answer: We appreciate the reviewer's insightful comments on investigating additional ring sizes. During the conceptual and initial stages of our optimization, we have considered other aliphatic ring sizes for the backbone. The synthesis approach for the thiazolium salts requires the corresponding cyclic ketones as starting materials. Notably and not obviously, the availability and costs of these ketones represents significant limitation.

- Cycloheptanone: 25 g –41 USD
- Cyclooctanone: 25 g –26 USD
- Cyclononanone: 1 g – 392 USD
- Cyclodecanone: 1 g –450 USD
- Cycloundecanone: 1 g – 1511 USD (not available in Switzerland)
- Cyclododecanone: 25 g –21 USD
- Cyclotridecanone: 1 g – 3250 USD (not available in Switzerland)
- Cyclotetradecanone: 1 g – 2893 USD (not available in Switzerland)

Besides reactivity and high selectivity, a dominant cornerstone of our ligand design principle is ease of synthetic access and affordability. All factors must be met and will later contribute to the acceptance and utilization of a new catalyst by the community. Due to the lack of availability and the prohibitively high costs, we categorically excluded C9, C10, C11, C13 and C14 ketone. Reported synthesis also involves tedious and low-yield processes. Due to the use in commercial routes to musk fragrances, the C12 ketone is the rare exception. The optimized reaction conditions with the 12-membered ring NHC catalyst demonstrate high yields and excellent enantioselectivity across all substrates. Along with availability and cost considerations, we prioritized this specific ring sizes in our development. Even in a possible event a larger ring would lead to minute improved selectivities, it would never become the go-to catalyst for the community.

Comment 2. For the comment 7 about the calculation study in light of the author's explanation, I can understand why the author did not calculate the energy gap between different transition states for the enantio-determining step. However, the DFT results presented in the current paper are somewhat rudimentary. Despite the vast conformational space of the organocatalyst and its catalytic intermediates, there is no mention of any attempts to verify the assignment of key structures as global conformational minimum. The process for geometry optimization for the NHC catalyst and the radical intermediates should be detailed. In addition, considering the solvent

significantly impacts the reaction, the presented calculation study, which was conducted in vacuo, should ideally incorporate solvent effects for enhanced accuracy.

Our answer: We are thankful for the reviewer's critique. We decided not only to address the reviewer's comment, but bring the computational part to the next level by including all possible diastereomeric transition state. Therefore, we have initiated a collaboration with computation chemistry specialists. The demanding calculations are highly complex and required significant computational power due an accumulation of several complicating factors:

- a) very high precision needed to account for small energy differences in the diastereotopic transition states
- b) large catalysts structures with a high number of atoms;
- c) highly flexible molecules with multiple rotatable bonds (backbone ring, four t-butyl groups, chiral side arms;
- d) two open shell intermediates.

We have now included a more thorough computational analysis in the manuscript. We began by running an extensive conformational search using the Crest software. Of the many thousands of conformers produced during this screening, we used an in-house developed conformer clustering tool (marc) to identify the 10 most distinct conformers lying within 15 kcal/mol of the global minimum identified by Crest. These structures were then subjected to full DFT optimizations (at the PBE0-D3(BJ)/def2-TZVP//PBE0-D3(BJ)/def2-SVP level including the SMD solvation model to account for solvent effects). From these DFT computations, the global minima for the organocatalyst NHC7, the ketyl radical of NHC7, as well as the four unique transition states $\Delta\Delta G^\ddagger$ (re-re, si-si, re-si, and si-re) leading to the major and minor enantiomer of the product were identified. In agreement with experimentally observed dominant (*S*)-enantiomer, our computations predict a similar enantiomeric ratio of 80:20 (*S*:*R*). We are confident that this substantial reinforcement on the computational aspect revisions further validates our finding and is very helpful for any further experimental application of our catalyst design.

Comment 3. We highly suggest the author should clarify this catalyst design from Li's group before ref. 47 in Figure 1D.

Our answer: We clarified the contribution from Li: "Li previously reported a simple member of this NHC catalyst family for an intramolecular C-H acylation reaction⁴⁷ and most recently applied the same catalyst to a 1,2-boron migrative acylation, again observing a moderate enantiomeric excess.⁶⁴"

Point-to-Point Answer

Reviewer #3 (Remarks to the Author):

The authors have well answered the questions, and managed to provide detailed DFT calculations to explain for the enantioselectivity generated. However, one key question is still remained to be uncertain: whether the reaction mechanism is a radical chain process or not. A radical-radical C-C coupling is not kinetically favored because of the low concentration of highly-reactive radical species. The benzyl radical E may attack the aldehyde **1a**, which is in a higher concentration, rather than the radical anion D. This is a shared challenge for both the present work and Li's non-enantioselective work. I think the answer to this question will help a lot in this field. Therefore, major revision is still needed.

Our answer:

We thank the reviewer for this valuable feedback. The question of whether the illustrated process is a radical-radical coupling or radical chain process is relevant. We can easily answer this last remaining query twofold:

1) As the reviewer mentioned, there are studies reported by the Li where, in most cases, carbon-centered radical-radical coupling has been proposed to form α -branched products. These mechanisms are widely accepted by the scientific community. Our proposed mechanism is built upon these prior studies and focuses specifically on computational analysis and the development of a catalytic model for the enantio-determining transition state. Several additional relevant reports for asymmetric reactions in the literature provide context supporting the mechanism:

Nature **2024**, 625, 74–78: [DOI:10.1038/s41586-023-06822-x] (**ref 23**)

Nature **2024**: [DOI:10.1038/s41586-024-08399-5] (recently published)

Angew. Chem. Int. Ed. **2023**, 62, e202312829 (**ref 24**)

While the first two studies focus on enzymatic catalysis, the underlying reaction mechanism is analogous. Additionally, ref 24 supports the same mechanism observed in our work. For this NHC-catalyzed reaction, radical-radical coupling leading to the formation of a new carbon-carbon bond has been well-recognized by the community and aligns with our findings. The type of mechanism is now very common and frequently dubbed «*radical sorting*».

2) For small molecule NHC catalysis, our highly enantioselective method provides a robust experimental evidence for the radical-radical coupling and against the radical chain mechanism. If as suggested by the reviewer a benzylic radical E attacks aldehyde **1a**, the C-C bond forming and enantiodetermining event would happen in the *absence* of the chiral environment of the catalyst inevitably leading to a racemic product. However, since our results demonstrate very high levels of enantioselectivity, the new C–C bond-forming event must be governed by the chiral environment. In our system, this is achieved through radical-radical coupling between radical D and radical E.